# Excess cases of influenza and the coronavirus epidemic in Catalonia: a time-series analysis of primary-care electronic medical records covering over 6 million people

Ermengol Coma Redon,[1,2] Nuria Mora,[1,2] Albert Prats-Uribe [ID] ,[3]
Francesc Fina Avilés,[1,2] Daniel Prieto-Alhambra [ID] ,[2,3] Manuel Medina[1,2]

ECR, NM and AP-U are joint first authors.

¹Sistemes d'Informació dels Serveis d'Atenció Primària (SISAP), ICS, Barcelona, Catalunya, Spain
²IDIAP Jordi Gol, Universitat Autònoma de Barcelona, Barcelona, Catalunya, Spain
³Nuffield Department of Orthopaedics Rheumatology and Musculoskeletal Science, University of Oxford, Oxford, UK

**Correspondence to**
Dr Daniel Prieto-Alhambra;
daniel.prietoalhambra@ndorms.ox.ac.uk

## ABSTRACT

**Objectives** There is uncertainty about when the first cases of COVID-19 appeared in Spain. We aimed to determine whether influenza diagnoses masked early COVID-19 cases and estimate numbers of undetected COVID-19 cases.

**Design** Time-series study of influenza and COVID-19 cases, 2010–2020.

**Setting** Primary care, Catalonia, Spain.

**Participants** People registered in primary-care practices, covering >6 million people and >85% of the population.

**Main outcome measures** Weekly new cases of influenza and COVID-19 clinically diagnosed in primary care.

**Analyses** Daily counts of both cases were computed using the total cases recorded over the previous 7 days to avoid weekly effects. Epidemic curves were characterised for the 2010–2011 to 2019–2020 influenza seasons. Influenza seasons with a similar epidemic curve and peak case number as the 2019–2020 season were used to model expected case numbers with Auto Regressive Integrated Moving Average models, overall and stratified by age. Daily excess influenza cases were defined as the number of observed minus expected cases.

**Results** Four influenza season curves (2011–2012, 2012–2013, 2013–2014 and 2016–2017) were used to estimate the number of expected cases of influenza in 2019–2020. Between 4 February 2020 and 20 March 2020, 8017 (95% CI: 1841 to 14 718) excess influenza cases were identified. This excess was highest in the 15–64 age group.

**Conclusions** COVID-19 cases may have been present in the Catalan population when the first imported case was reported on 25 February 2020. COVID-19 carriers may have been misclassified as influenza diagnoses in primary care, boosting community transmission before public health measures were taken. The use of clinical codes could misrepresent the true occurrence of the disease. Serological or PCR testing should be used to confirm these findings. In future, this surveillance of excess influenza could help detect new outbreaks of COVID-19 or other influenza-like pathogens, to initiate early public health responses.

### Strengths and limitations of this study

► We used good quality data covering >6 million people and >85% of the Catalan population, obtained directly from primary record records.
► Data had previously been validated against gold-standard influenza sentinel systems.
► We used ecological data and modelled it using data from previous seasons, therefore assuming a direct link between excess influenza cases and the COVID-19 pandemic.
► Excess influenza cases could also have been due to a panic effect, where current coronavirus epidemic had encouraged people to consult healthcare professionals more frequently and for milder symptoms than usual.
► We lack confirmatory tests or antigenic data for the estimated excess influenza cases, but our results agree with the proportion of influenza samples that tested positive for SARS-CoV-2 in a recent study.

## BACKGROUND

A new infectious disease, now named COVID-19, was identified by Chinese authorities on 7 January 2020 as the cause of an outbreak of pneumonia in Wuhan.[1] Caused by SARS-CoV-2, COVID-19 is asymptomatic or presymptomatic in a high proportion of patients, with estimates around 15%–30%.[2–4] Most patients present mild influenza-like symptoms, including fever, dry cough, fatigue, sore throat, dyspnoea, headache and myalgia.[5 6] Around 15%–20% of symptomatic cases present severe forms of disease that require hospital admission.[1 7] Older people, men and those with multiple comorbidities appear more likely to suffer more serious types of COVID-19.[5 6 8–10] Conversely, children seem to have a similar probability of

infection, but milder and often asymptomatic forms of the disease.[11]

Cases of COVID-19 have grown exponentially and have been reported all over the world. The first three cases in Europe were reported in France on 24 January 2020.[12] The first imported COVID-19 case in Spain was dated 31 January 2020 in La Gomera, and the first in Catalonia reported a month after, on 25 February 2020. The total number of confirmed cases in Catalonia then increased exponentially, with 715 cumulative cases reported by 14 March 2020 and a striking 4203 on 20 March 2020. Despite these official figures, it is uncertain whether SARS-CoV-2 was circulating in the community before the first official cases. It is difficult to believe, for example, that this airborne infection did not cross the uncontrolled borders between Catalonia and France for a whole month. Some have thus speculated that undetected COVID-19 cases may have been categorised as influenza before the first official case was reported in Spain.[13]

Catalonia is fortunate to have a reliable system for influenza surveillance in place. A network of 60 sentinel general practitioners (GPs) covering 1% of the total population report daily cases of influenza-like illness (ILI) and take samples for differential diagnosis and confirmation of influenza infections in the region.[14] A specialised hospital-based system takes samples from severe hospitalised influenza cases.[14] A community-based surveillance system called Diagnosticat also extracts counts of ILI diagnoses from a network of GP health records in real-time, covering 85% of the population.[15] This last approach allows us to examine trends with granularity and to stratify analyses by age and other factors.

As the first cases of SARS-CoV-2 appeared in Catalonia during the influenza epidemic season and the disease shares some symptomatology with influenza, we hypothesised that SARS-CoV-2 could have been circulating in the community before the first confirmed case, resulting in an excess of influenza diagnoses. We aimed to estimate the number of excess influenza cases in Catalonia, globally and by age, and to examine its relationship with the number of clinically diagnosed COVID-19 cases.

## METHODS

We used a time-series study of influenza and COVID-19 cases. We extracted data from primary-care electronic medical records covering about 85% of the population of Catalonia, around 6 million people. The study period included all influenza seasons from autumn–winter 2010–2011 to autumn–winter 2019–2020.

The key study outcomes were diagnoses of influenza and COVID-19. Daily frequency of influenza cases recorded in primary-care records were obtained from electronic medical records, as is routinely done for the Diagnosticat database.[16]

Diagnosticat is a website that reports in real-time all influenza diagnoses recorded by all GPs working at any of the primary-care centres run by the Institut Català de la Salut (ICS). ICS is the main primary-care health service provider in Catalonia and covers about 85% of practices in the region, who all use the same electronic medical record software, ECAP.[17] Diagnosticat includes all clinical influenza diagnosis codes (International Classification of Diseases-10 codes in online supplementary table 1) and is updated from ECAP daily (since 2010). It presents the frequency of daily influenza cases and the weekly incidence rates per $10^5$ population, a unit that allows diagnoses to be compared between territories independently of the number of inhabitants. Influenza data on Diagnosticat has been shown to accurately represent that in a gold-standard source, the sentinel network of influenza infection reports data set.[15]

The number of COVID-19 clinical diagnoses were extracted and aggregated using the same data source and methods as for influenza diagnoses. Clinical diagnoses of COVID-19 have been recorded in ECAP since 27 February 2020, when bespoke codes were introduced (online supplementary table 1). Since 15 March 2020, Catalan policies have advocated for cases to be defined based on symptoms alone, with serological or PCR confirmation only required when patients are admitted to hospital or are healthcare staff.[18]

### Statistical analysis

Daily counts of influenza and COVID-19 cases were computed based on the frequency of cases recorded in the previous 7-day period to avoid weekly effects on recording practice. All influenza seasons in the study period (2010–2011 to 2019–2020) were analysed separately to characterise annual epidemic curves for seasonal influenza.

Influenza seasons with a visually similar epidemic curve and similar peak case number to that of the 2019–2020 season were selected to model predictions for 2019–2020. We selected these specific seasons after an assessment of the number of cases at the peak to maximise comparability with the current influenza season before the COVID-19 outbreak. Auto Regressive Integrated Moving Average (ARIMA) models[19] were fitted to the seasons included in the analysis for the whole population and for three age groups, paediatric patients (under 15), adults (15–64) and elderly (over 64 years old).

From the fitted time series, the expected speed of decrease in the number of weekly influenza cases for the 2019–2020 influenza season was calculated for each day after the peak. The expected speed of decrease was defined as the difference between the number of influenza diagnoses predicted between the current day t and the previous day t−1, divided by the number of diagnoses predicted for the previous day t−1 ($(cases_t (cases_{t-1}) - 1)$). Expected influenza cases were calculated using the sequence $G_t = G_0 * \prod_{k=1}^{t} V_k$, where $G_t$ was the expected influenza cases in the period t, $G_0$ the number of cases at the peak and $V_k$ the speed of decrease at day k.

The expected influenza cases for each day on the 2019–2020 season were calculated from the day of the season

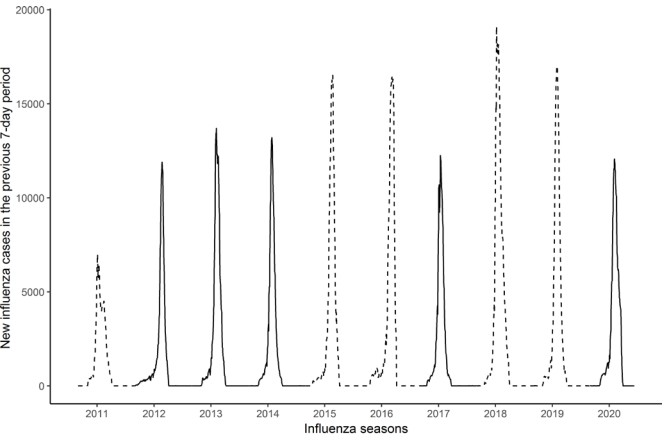

**Figure 1** Epidemic curves showing the weekly number of new influenza cases during the influenza seasons from autumn–winter 2010–2011 to autumn–winter 2019–2020 in Catalonia, Spain. Curves in solid lines were similar to the 2019–2020 season and included in further modelling. Curves in dashed lines were not similar to the 2019–2020 season and were excluded from further modelling.

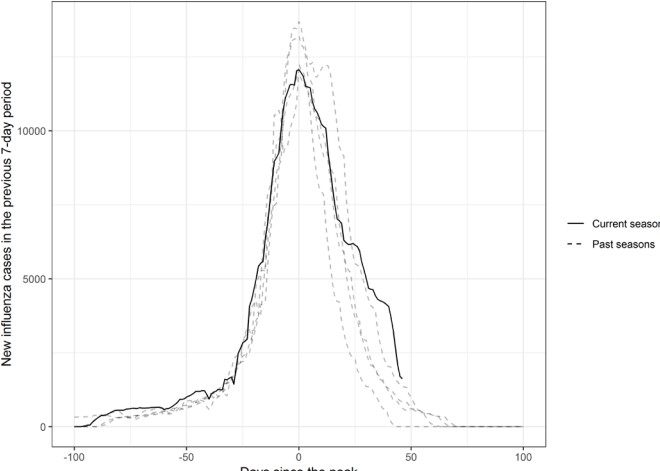

**Figure 2** Epidemic curves for the 2019–2020 Catalonia influenza season (solid line) and the four seasons in the past decade with a similar peak number of cases (dotted lines: 2011–2012, 2012–2013, 2013–2014 and 2016–2017), centred on the day of the peak number of cases in each curve.

peak to 20 March 2020, the day of the data extraction. Excess influenza cases were defined as the number of observed minus expected cases, estimated daily as above. We calculated 95% CIs for each estimate. All analyses were performed in R V.3.5.1.[20]

We further tested our method, as a sensitivity analysis, with data from the most recent (2018–2019) season as a negative control. We checked whether the method was able to identify the season as a 'regular' influenza season not detecting excess influenza cases.

### Patient and public involvement

This research was done without patient involvement. Patients were not invited to comment on the study design and were not consulted to develop patient relevant outcomes or interpret the results. Patients were not invited to contribute to the writing or editing of this document for readability or accuracy.

### RESULTS
### Previous influenza epidemic curves

Four of the previous nine influenza season curves (2011–2012, 2012–2013, 2013–2014 and 2016–2017) had an epidemic curve and number of influenza cases during the peak similar to the 2019–2020 season, as shown in figure 1. These four curves were used to estimate the number of expected cases of influenza in 2019–2020. The mean peak number of cases in the included and excluded seasons was 12 762 and 14 680, respectively. The peak number of cases in 2019–2020 was 12 066.

ARIMA models were fitted using the included seasons. Online supplementary table 2 shows the full modelling process and the fitted parameters.

### 2019–2020 influenza epidemic description

In Catalonia, the 2019–2020 influenza epidemic reached its peak on 4 February 2020, with 12 066 cases in the previous 7 days. Figure 2 shows the evolution of the season compared with past seasons, centred on the day of the peak. By eye, the downwards trend after the peak initially looks very similar to the previous seasons. However, 20 days after the peak, the curve starts to flatten, and the slope slows down. This abnormal pattern in the descending part of the curve differs from the pattern in the previous seasons.

### Expected versus observed cases

Figure 3 shows the observed and estimated numbers of weekly new influenza cases (with 95% CI) after the peak of the 2019–2020 influenza season. The estimated expected number of cases were predicted using the selected previous influenza seasons.

In the whole population, observed cases were greater than expected after the seasonal influenza peak, but not always significant during the whole study period. The difference was statistically significant for 23 days between 4 February 2020 and 20 March 2020. Most of these days fell after 8 March 2020, when the difference between observed and expected increased significantly and observed cases remained above the 95% CI band for expected cases for 2 weeks.

There was a greater difference between observed and expected cases among people aged 15–64 years than in both the total population and other age groups, with 25 total days of significant difference. The observed and expected cases diverged earlier than for the total population, separating around 26 February 2020 and remaining significantly different for the rest of the study period.

Observed and expected cases were generally similar in those older than 64 years, until 6 March 2020. Observed cases then quickly rose above expected cases, with the

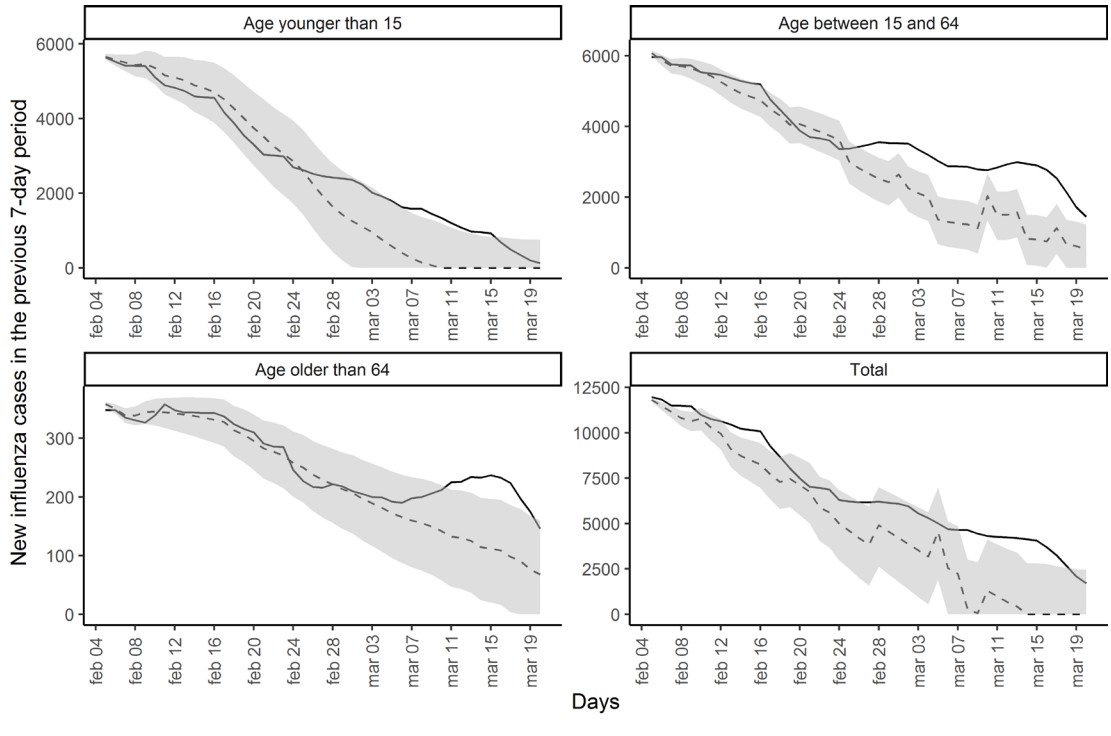

**Figure 3** Observed and expected (with 95% CI) weekly new influenza cases each day after the peak of the 2019–2020 Catalonia influenza season, in the full population and in each age group.

difference becoming significant on 11 March 2020 and remaining so for 9 days, until 19 March 2020.

The shape of the observed cases curve for people younger than 15 years was similar to that for people aged 15–64 years. However, the difference between observed and expected cases was only significantly different for 11 days, between 6 March 2020 and 16 March 2020.

We estimated 8017 excess influenza cases (95% CI: 1841 to 14 718) between 4 February 2020 and 20 March 2020. This excess is presented stratified by age in table 1.

Results for our negative control influenza season are shown in online supplementary figure 1. We found no excess influenza in the previous (2018–2019) influenza season using the same method.

### Excess influenza cases compared with COVID-19 diagnoses

Figure 4 depicts the number of excess influenza cases and COVID-19 diagnoses each day after the peak of the 2019–2020 seasonal influenza epidemic. Excess influenza cases increased rapidly from 24 February 2020, peaking on 7 March 2020. They steeply declined from 15 March 2020, coinciding with an increase in the number of COVID-19 diagnoses.

There were 4347 excess influenza cases and 1497 clinical diagnoses of COVID-19 on 14 March 2020, comparing with just 2575 excess influenza cases (40% less) and a striking 16 547 (539% increase) clinical diagnoses of COVID-19 on 20 March 2020.

### DISCUSSION

In mid-February 2020, we observed an unusually high, larger than expected number of influenza cases in the daily published data. In Catalonia, the 2019–2020 seasonal influenza epidemic reached its peak on 4 February 2020.

**Table 1** Number of excess influenza cases in Catalonia from 4 February 2020 to 20 March 2020, after the peak of the seasonal influenza epidemic, and the percentage of all influenza cases in that period that they make up, overall and by age group

| Age group | Estimated number of excess influenza cases (95% CI) | Percentage of all influenza cases in this age group made up by the estimated excess cases (95% CI) |
|---|---|---|
| Younger than 15 | 2078 (160 to 4078) | 13.6 (1.0 to 26.7) |
| Between 15 and 64 | 4670 (2387 to 7124) | 20.9 (10.7 to 31.8) |
| Older than 64 | 142 (33 to 260) | 8.9 (2.1 to 16.3) |
| Total | 8017 (1841 to 14 718) | 20.4 (4.7 to 37.5) |

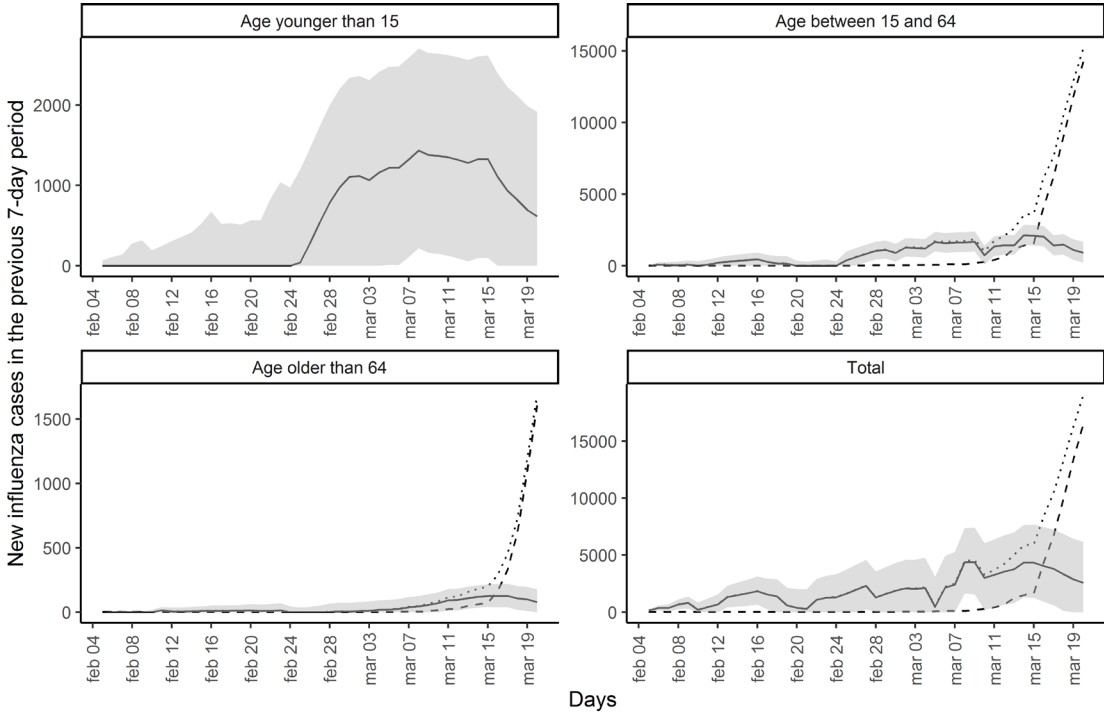

**Figure 4** Excess influenza cases and clinically diagnosed COVID-19 cases in Catalonia, Spain, as number of cases in the previous 7-day period, from the peak of the 2019–2020 seasonal influenza epidemic (4 February 2020).

Based on previous years' data, influenza diagnoses were expected to decrease rapidly over the following weeks. However, the number of influenza diagnoses instead remained stable, which was counterintuitive and inconsistent with data from past influenza seasons. This increase in observed influenza diagnoses over those expected, here named 'excess influenza', correlates over time with the observed number of COVID-19 cases. Excess influenza cases could be used in future for the early detection of competing outbreaks.

Using four of the previous nine influenza seasons as a benchmark, we detected 8017 excess influenza cases between 4 February 2020 and 20 March 2020. This excess was higher in people aged 15–64 years, with over 20% more cases than expected. The excess started to decrease after 15 March 2020. Worryingly, these results suggest that SARS-CoV-2 could have already been circulating in the Catalan population when the first imported case was reported on 25 February 2020. People infected with COVID-19 may have been masked under ILI diagnoses in primary care, allowing continuing community transmission of COVID-19 before public health measures were taken.

To our knowledge, this is the first study attempting to quantify the start of the COVID-19 epidemic in Spain by comparing the number of reported ILI cases with the expected figures based on previous influenza seasons. The excess influenza cases metric could be useful for monitoring future outbreaks of COVID-19 and other competing viral epidemics.

Our study has several limitations. We used ecological data and modelled it using data from previous seasons, therefore assuming a direct causal link between excess influenza cases and the COVID-19 pandemic. As our method is based on crude count of influenza cases, major changes in denominator and population structure could limit the use of the proposed method, but population age and gender has remained relatively stable in the study period.[21] Our main limitation is the possible misclassification of disease status due to limitations related to the use of clinical codes. We lack serological tests or antigenic data for confirmation, and this should be investigated to confirm our findings. Our results agree with a study that tested all influenza samples in Los Angeles for SARS-CoV-2, finding 2.2%–10.7% of the tested samples positive for the pathogen, and a centre for disease control report that times the start of limited community transmission round mid-January to February.[22 23]

The observed excess influenza cases could have been due to a panic effect, in which the current coronavirus infodemic, a rapid spread of misinformation, has encouraged people to consult healthcare professionals more frequently and for milder symptoms than usual. However, our data showed that the number of influenza diagnoses dropped drastically and COVID-19 diagnoses increased after 15 March 2020. New COVID-19 guidelines were released on 15 March 2020 in Spain that recommended only testing hospital-admitted patients and healthcare staff and encouraging GPs to diagnose COVID-19 clinically without PCR confirmation.[18] At least some of the

excess ILI cases were thus likely to have actually been COVID-19 cases.

Our study also has strengths. The data used were good quality, as demonstrated in many previous publications,[24–30] were obtained directly from primary-care records,and have been validated against gold-standard sentinel systems. This existing database covers over 85% of the population of Catalonia, which allowed us to rapidly detect excess influenza cases across the whole population and in different age groups.

In conclusion, the full extent of the COVID-19 pandemic is still unknown. The confirmed number of cases may be just the tip of the iceberg, due to the lack of testing of patients presenting mild COVID-19 symptoms. We need comprehensive, well-designed, seroprevalence studies to know how many people have been infected. This novel analysis approach could offer a quantitative approach to population surveillance that may be useful for other institutions/regions/countries and could be easily integrated into current information systems. This surveillance of excess influenza cases using widely available primary-care electronic medical records could help detect new outbreaks of COVID-19 and other ILI-causing pathogens, supporting early testing and public health responses.

**Acknowledgements** The authors acknowledge English language editing by Dr Jennifer A de Beyer of the Centre for Statistics in Medicine, University of Oxford.

**Contributors** ECR, NM, AP-U, FFA, MM and DP-A contributed to the design of the study, the interpretation of the results, and reviewed the manuscript. EC and NM had access to the data, performed the statistical analysis, and acted as guarantors. ECR, NM, AP-U, and DP-A are joint first authors and wrote the first draft of the manuscript. MM and DP-A are joint senior authors.

**Funding** The research was partially supported by the National Institute for Health Research (NIHR) Oxford Biomedical Research Centre (BRC). DP-A is funded through an NIHR Senior Research Fellowship (Grant number SRF-2018-11-ST2-004). The views expressed in this publication are those of the author(s) and not necessarily those of the NHS, the National Institute for Health Research or the Department of Health. AP-U is supported by Fundacion Alfonso Martin Escudero and the Medical Research Council (grant numbers MR/K501256/1, MR/N013468/1).

**Competing interests** DP-A reports grants and other from AMGEN; grants, non-financial support and other from UCB Biopharma; grants from Les Laboratoires Servier, outside the submitted work; and Janssen, on behalf of IMI-funded EHDEN and EMIF consortiums, and Synapse Management Partners have supported training programmes organised by DP-A's department and open for external participants. AP-U reports grants from Fundacion Alfonso Martin Escudero and the Medical Research Council. No other relationships or activities that could appear to have influenced the submitted work.

**Patient and public involvement** Patients and/or the public were not involved in the design, or conduct, or reporting or dissemination plans of this research.

**Patient consent for publication** Not required.

**Provenance and peer review** Not commissioned; externally peer reviewed.

**Data availability statement** Data are available upon reasonable request. Data on flu cases are publicly available. All other aggregated data are available upon reasonable request.

**ORCID iDs**
Albert Prats-Uribe http://orcid.org/0000-0003-1202-9153
Daniel Prieto-Alhambra http://orcid.org/0000-0002-3950-6346

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
