## [Reviewer comments · BMJ Open]

ARTICLE DETAILS

TITLE (PROVISIONAL)	Excess cases of influenza and the coronavirus epidemic in Catalonia: a time series analysis of primary care electronic medical records covering over 6 million people
AUTHORS	Coma Redon, Ermengol; Mora, Nuria; Prats-Urbe, Albert; Fina Avilés, Francesc; Prieto-Alhambra, Daniel; Medina, Manuel

VERSION 1 – REVIEW

REVIEWER	Patrick Ryan Janssen Research and Development, USA PBR is employee of Janssen Research and Development and shareholder of Johnson & Johnson
REVIEW RETURNED	22-Apr-2020

GENERAL COMMENTS	The team has provided a nice population summary of diagnosis trends to compare the existing COVID pandemic with prior influenza seasons, to suggest that misdiagnosis of COVID may have been observed prior to when cases were initially recognized. This is highly pertinent to the global effort where case identification is critically important for managing the disease, and it would be a valuable exercise to see conducted in other geographies. (A related and timely aside, but just today in NYTimes, there is story on how death re-classification suggests the disease was in California, US, weeks prior to first case: https://www.nytimes.com/2020/04/22/us/coronavirus-first-united-states-death.html) I would suggest the authors specifically mention that this novel analysis approach could offer a quantitative approach to population surveillance that may be useful for other institutions/regions/countries. The limitation is mentioned clearly but it could be further reinforced, perhaps even in the abstract, that the analysis is focused on diagnosis codes (which represents clinical (mis) perception of patient presentation) and does not reflect true disease status, as could be theoretically measured by serology. How would diagnostic coding be influenced over time by changes in policies around lab testing, or by patients' awareness of the symptoms to report to clinicians?. A minor comment, but it caught my eye as a reader, was that prior flu seasons were not consistently selected (e.g. 2014-2015, 2015-2016, 2017-2018 excluded, because the number of cases during peak were higher, what is the implication of that?), that could be explained just to pacify the curiosity that'll be raised when readers see the 'excluded' curves in Figure 1, but I do not think it legitimately detract from the overall effort. Given the model is predicated on case counts, it would be useful to further clarify the population demographics and that you expect the catchment is stable over time. From a methodological perspective, can the prior flu seasons be used as negative/positive controls to
--

	demonstrate that method who not have found anything alarming in prior years, and only the 2019-2020 exhibits an interesting pattern?
--	--

REVIEWER	Rebekah Stewart CDC, USA
REVIEW RETURNED	08-Jun-2020

GENERAL COMMENTS	Thank you for the opportunity to review your manuscript. This is a timely paper, with high-impact potential, using a strong data source. This paper could be strengthened by more clearly defining some of the methods and by referencing the most up-to-date and accurate literature for COVID-19. Please see below for a more detailed review. Strengths  - Large dataset covering >6 million people and >85% of the Catalan population - Estimated case numbers for influenza were predicted using four previous influenza seasons Weaknesses  - Cases of both influenza and COVID-19 diagnosed clinically Background  - Page 5, line 9. The authors cite a WHO mission report stating that COVID-19 is asymptomatic in around 1-3% of patients. This datapoint is out of date at best. There are several studies that suggest the proportion of people with asymptomatic or presymptomatic COVID-19 is much higher, perhaps as high as 50%. Please see below for a few references, however, there are many more which could be cited:  o Mizumoto et al. 2020. Estimating the Asymptomatic Proportion of COVID-19 Cases on Board the Diamond Princess. Euro Surveill o Kimball et al. 2020. Asymptomatic and Presymptomatic SARS-CoV-2 Infections in Residents of a Long-Term Care Skilled Nursing Facility. MMWR o Buckley et al. 2020. Universal testing of patients and their support persons for COVID-19 when presenting for admission to labor and delivery within the Mount Sinai Health System. AJOG. - Page 5, line 14. There are several other research papers that could be cited for this statement. - Page 5, line 33. Suggest naming the “northern neighboring country” that is referenced in this sentence. Methods  - Most are clearly described - I recognize that ICD-10 codes were used to differentiate between influenza and COVID-19 diagnoses, but methods would be improved with an explanation of the case definition being used by clinicians for both influenza and COVID-19. - Missing threshold for statistical significance Results  - Page 8, line 25. Please revise this sentence. Seems contradictory as written as the authors state that observed cases were always greater than expected, to some extent. - Is there any possibility of going back and testing any laboratory samples during the initial time period of excess influenza cases? A PCR-positive test for SARS-CoV-2 during that time period would strengthen your findings substantially. Discussion  - Page 10, line 35. Consider adding this reference to your discussion: CDC COVID-19 Response Team. 2020. Evidence for
---

	Limited Early Spread of COVID-19 within the United States, January-February 2020. MMWR. - Page 10, line 59. While the pathogen is SARS-CoV-2, I believe the pandemic should be referred to as the disease, COVID-19
--	---

VERSION 1 – AUTHOR RESPONSE

Reviewer 1 - Patrick Ryan

The team has provided a nice population summary of diagnosis trends to compare the existing COVID pandemic with prior influenza seasons, to suggest that misdiagnosis of COVID may have been observed prior to when cases were initially recognized. This is highly pertinent to the global effort where case identification is critically important for managing the disease, and it would be a valuable exercise to see conducted in other geographies. (A related and timely aside, but just today in NYTimes, there is story on how death re-classification suggests the disease was in California, US, weeks prior to first case: <https://www.nytimes.com/2020/04/22/us/coronavirus-first-united-states-death.html>)

1- I would suggest the authors specifically mention that this novel analysis approach could offer a quantitative approach to population surveillance that may be useful for other institutions/regions/countries.

- Thank you for this comment. We have now added a statement along these lines in the Discussion: “This novel method could offer a quantitative approach to population surveillance that may be useful for other institutions/regions/countries and could be easily integrated into current information systems”

2- The limitation is mentioned clearly but it could be further reinforced, perhaps even in the abstract, that the analysis is focused on diagnosis codes (which represents clinical (mis) perception of patient presentation) and does not reflect true disease status, as could be theoretically measured by serology.

- This is a very good point. We have added the following statement in the abstract: “The use of clinical codes in the current study could misrepresent the true occurrence of the disease. Serological or PCR testing should be used to confirm these findings.”

- We have also added the following sentence in page 10: “Our main limitation is the possible misclassification of disease status due to limitations related to the use of clinical codes. We lack serological tests or antigenic data for confirmation, and this should be investigated to confirm our findings”

3- How would diagnostic coding be influenced over time by changes in policies around lab testing, or by patients' awareness of the symptoms to report to clinicians?.

- This is a point for concern for longer term and future studies. However, the current data includes a short period when testing was only allowed for hospitalised patients or health professionals. We therefore believe this should not be a major limitation of our study.

4 - A minor comment, but it caught my eye as a reader, was that prior flu seasons were not consistently selected (e.g. 2014-2015, 2015-2016, 2017-2018 excluded, because the number of cases during peak were higher, what is the implication of that?), that could be explained just to pacify the curiosity that'll be raised when readers see the 'excluded' curves in Figure 1, but I do not think it legitimately detract from the overall effort.

- Thank you for this comment, which has also been usefully raised by others during the preprint 'life' of our manuscript. We selected these specific seasons after an assessment of the number of cases at the peak to maximise comparability with the current flu season before the covid19 outbreak. We have now clarified this in the Methods section of the paper: “We selected these specific seasons after an assessment of the number of cases at the peak to maximise comparability with the current flu season before the COVID-19 outbreak.”

5 - Given the model is predicated on case counts, it would be useful to further clarify the population

demographics and that you expect the catchment is stable over time.

- Thank you for pointing out this limitation. Official figures for Catalonia show a relatively stable population for the region in the study period (link). We therefore believe this should not have a great impact on our findings. We have however added a statement in the Limitations section: “As our method is based on crude count of flu cases, major changes in denominator and population structure could limit the use of the proposed method, but population age and gender has remained relatively stable in the study period.”

6 - From a methodological perspective, can the prior flu seasons be used as negative/positive controls to demonstrate that method who not have found anything alarming in prior years, and only the 2019-2020 exhibits an interesting pattern?

- We thank the reviewer for this suggestion. We agree that this analysis will improve the quality of our manuscript. We tested our method with data from the most recent (2018-2019) season, with results shown in the following figure:

- Our method did not find any relevant departure in the 2018-2019 season, hopefully helping the validation of our method. We have now added this information to the manuscript in the “Methods” and “Results” sections.

- Methods: “We further tested our method, as a sensitivity analysis, with data from the most recent (2018-2019) season as a negative control. We checked whether the method was able to identify the season as a “regular” flu season not detecting excess influenza cases.”

- Results: “ Results for our negative control flu season are shown in Supplementary Figure 1. We found no excess influenza in the previous (2018-2019) flu season using the same method.”

Reviewer 2 - Rebekah Stewart

Page 5, line 9. The authors cite a WHO mission report stating that COVID-19 is asymptomatic in around 1-3% of patients. This datapoint is out of date at best. There are several studies that suggest the proportion of people with asymptomatic or presymptomatic COVID-19 is much higher, perhaps as high as 50%. Please see below for a few references, however, there are many more which could be cited:

- Apologies, a lot has changed in the last 1-2 months since we wrote this paper. Thank you for proposing new references, we have changed the references and reworded the statement in the revised manuscript: “is asymptomatic or presymptomatic in a high proportion of patients, with estimates around 15 to 30%.”

Page 5, line 14. There are several other research papers that could be cited for this statement.

- Thank you for this comment. We have added several references for this statement.

Page 5, line 33. Suggest naming the “northern neighboring country” that is referenced in this sentence.

- We agree and have changed the phrasing of that sentence, page 5 line: “northern neighboring country” for “France”

I recognize that ICD-10 codes were used to differentiate between influenza and COVID-19 diagnoses, but methods would be improved with an explanation of the case definition being used by clinicians for both influenza and COVID-19.

- Clinical practice for the diagnosis of influenza and COVID-19 is relatively homogeneous in the study period, but probably differs by practising physician. The current study included over 3,000 primary care practitioners, therefore limiting our ability to ascertain this with sufficient granularity.

- We have however added a statement in the limitations, as also suggested by reviewer 1, to clarify that validation of our findings using laboratory testing would be a good next step to further validate our method.

Missing threshold for statistical significance

- We have added a sentence to specify the confidence intervals in the Methods section: “We calculated 95% confidence intervals for each estimate.”

Page 8, line 25. Please revise this sentence. Seems contradictory as written as the authors state that observed cases were always greater than expected, to some extent.

- We have changed the sentence to make it clearer in the Results: “In the whole population, observed cases were greater than expected after the seasonal influenza peak, but not always significant during the whole study period. “

Is there any possibility of going back and testing any laboratory samples during the initial time period of excess influenza cases? A PCR-positive test for SARS-CoV-2 during that time period would strengthen your findings substantially.

- We realise this is one of the limitations of this study. There is a new project from our institution that will perform antigenic testing, but we don’t have access to historical flu samples to definitely confirm our findings.

- We have therefore reinforced this limitation and we call for testing of historical samples and serological testing in the Discussion: “Our main limitation is the possible misclassification of disease status due to limitations related to the use of clinical codes. We lack serological tests or antigenic data for confirmation, and this should be investigated to confirm our findings.”, and in the Abstract; “The use of clinical codes in the current study could misrepresent the true occurrence of the disease. Serological or PCR testing should be used to confirm these findings.”

Page 10, line 35. Consider adding this reference to your discussion: CDC COVID-19 Response Team. 2020. Evidence for Limited Early Spread of COVID-19 within the United States, January-February 2020. MMWR.

- We have cited this report and added a sentence to the discussion in the Discussion “and a CDC report that times the start of limited community transmission round mid-January to February”

Page 10, line 59. While the pathogen is SARS-CoV-2, I believe the pandemic should be referred to as the disease, COVID-19

- We have applied the proposed change in the Discussion: “COVID-19”

VERSION 2 – REVIEW

REVIEWER	Patrick Ryan Janssen Research and Development, Titusville, NJ, USA Columbia University Medical Center, New York, NY, USA PBR is employee of Janssen Research and Development and shareholder in Johnson & Johnson.
REVIEW RETURNED	01-Jul-2020
GENERAL COMMENTS	The revisions nicely address all of the comments from the reviewers. I think this paper will provide a useful contribution and can be potentially important as we look toward the next few season and the potential to need to differentiate influenza from COVID-19.